# Same-Sex Parenting Competence Evaluation: The Role of Gender Essentialism, Political Orientation, and Attribution of Conflict

**Chiara Ballone** [1,*], **Maria Giuseppina Pacilli** [2,*], **Manuel Teresi** [1], **Alessandro Taurino** [3], **Daniele Paolini** [4] and **Stefano Pagliaro** [1]

1 Department of Neuroscience, Imaging and Clinical Sciences, University of Chieti-Pescara, 66100 Chieti, Italy
2 Department of Political Science, University of Perugia, 06123 Perugia, Italy
3 Department of Education, Psychology and Communication, University of Bari, 70121 Bari, Italy
4 Department of Human Science, Italian University Line, 50122 Florence, Italy
* Correspondence: chiara.ballone@unich.it (C.B.); maria.pacilli@unipg.it (M.G.P.)

**Abstract:** Many countries are discussing LGBT rights, such as the right to wed and create a family. Despite research indicating that there is no reason to deny same-sex couples the right to be parents, negative attitudes persist concerning the quality of parenting by gay and lesbian individuals. The purpose of this study ($N = 436$) was to explore the relationship between the attribution of conflict in same-sex couples and the attribution of lower parenting competencies. We examined the attribution of conflict within heterosexual vs. same-sex couples in order to determine if the alleged conflict attributed to the latter can be used in a strategic manner to justify reduced same-sex parenting competence. Results showed a positive association between the attribution of conflict and lower parenting competence, especially in the same-sex couple evaluation. Furthermore, the attribution of conflict appears to be associated with a conservative political stance, gender essentialist beliefs, and homonegativity. A moderated mediation model confirmed our prediction, revealing that right-wing (vs. center and left-wing) participants considered same-sex couples to be less competent as parents due to the attribution of conflict within the couple. Results might be useful to foster the dissemination of reliable information about same-sex parent families.

**Keywords:** gender essentialism; attitudes towards same-sex parenting; gender-role reliefs; sexual prejudice; attribution of conflict

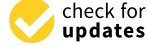



## 1. Introduction

In recent decades, the traditional understanding of what constitutes a family has drastically changed, adapting to consider several family configurations such as for instance single parent, reunited families, and same-sex parents (Petruccelli et al. 2015; Lingiardi and Carone 2016a). Specifically, same-sex parenting, commonly referred to as homoparenthood, is steadily growing despite the fact that it has always been viewed as a hidden occurrence.

In 2004, based on consistent and solid scientific evidence, the APA Council of Representatives adopted a policy resolution that stated that lesbian and gay parents are just as likely as straight parents to provide supportive and healthy environments for their children (American Psychological Association 2004). Nevertheless, the debate over children's growth in same-sex parent families remain a concern, especially in the Italian context (Taurino 2016). One of the numerous negative stereotypes about same-sex relationships is the belief that same-sex couples are more unstable than heterosexual ones due to the rumored promiscuity and an unsatisfactory bond between partners (Chiari and Borghi 2009). This prejudice is completely unsupported by evidence: research has shown that among the strengths of same-sex couples are good communication and negotiation skills, commitment, and egalitarianism in couple management (see Rostosky and Riggle 2017 for

a review). Moreover, further empirical evidence indicates that the negative effect of conflict on children's behavioral and adjustment problems occur regardless of family composition and parent's sexual orientation (Amato and Keith 1991; Chan et al. 1998; Flaks et al. 1995). Starting from this premise, in the present research we focused on the attribution of conflict within heterosexual couples and same-sex couples in order to investigate whether the alleged conflict attributed to same-sex couples vs. heterosexual couples can be used in strategic way to justify reduced same-sex parenting competence. Despite the fact that negative relation between attribution of conflict and parenting abilities exists beyond the sexual orientation of the partners, we advance that among those who endorse an ideology that embeds homophobic stances, that is among right wingers, such a relation can in fact be used to somewhat justify the beliefs that same-sex couples should not have the competences to raise a child. More specifically, we advance that the perception that conflict is primarily problematic only in same-sex relationships may be a further manifestation of prejudice against same-sex parent families. In Western countries, the right to form a family remains precarious for same-sex parents (Monaco 2022). In terms of parental competence and ability to provide a nurturing environment for their children, same-sex parents are subjected to deep scrutiny. As such, same-sex parent families are expected to exhibit hyper-appropriateness and perfection to avoid negative judgments and discrimination and to be ultimately socially accepted (Lingiardi and Carone 2016b).

### 1.1. Attitudes towards Same-Sex Parents

Unlike heterosexual couples, same-sex couples tend to make more effort to acquire "legitimate parental credentials" (Taylor 2009). The legitimacy of same-sex parenting appears to be hampered by a conservative viewpoint that views these opportunities as undermining the normative definition of family, which is a family rooted on natural rights (Green 2013). This idea is consistent with the heteronormative mold, which favors unions based on a dichotomy of gender where parental responsibilities are mostly linked to the biological sex (Biblarz and Stacey 2010). However, research comparing the functionality of the family and the well-being of children in same-sex and heterosexual parent families, have consistently shown the parental effectiveness of different family structures (Bos et al. 2005; Johnson and O'Connor 2002; Lelleri et al. 2008). Dysfunctional components might exist regardless of the structure of the family configuration and parenting consists of a capability entirely detached from the family structure (Fruggeri 2005). Therefore, academics have focused on the effectiveness of same-sex parenting, considering the health of children to be an essential outcome of the functioning of parenthood (Bos et al. 2005; Lelleri et al. 2008).

Although several studies have indicated the effectiveness of same-sex parent families showing that parents' skills are not related to sexual orientation (Baiocco et al. 2018; Bos et al. 2007; Bos and Sandfort 2010), there is a stigma surrounding same-sex parent families (Pistella et al. 2018; Costa 2022). A considerable portion of the public appears to be opposed to same-sex marriage and parenting, and the effects of this phenomenon on the child's growth continue to be the subject of dispute. The core issue is the belief that in a healthy family both men and women are required for parental responsibilities (Biblarz and Stacey 2010; Pacilli et al. 2011). This view is based on the belief that parental roles are inextricably tied to a person's biological sex (Biblarz and Stacey 2010) which means that adhering to tradition, which provides a family proper model, prevents us from approaching new family structures and leads us to dismiss same-sex parents, adoptive, and single parents from the parental spectrum. This bias is strongly tied to a violation of people's established gender role expectations (Capezza 2007; Petruccelli et al. 2015). Gender roles have been characterized as the existence of roles that seem to be more appropriate for men or women (Whitley 2001). Both men and women must adapt their duties and actions, as well as their physical characteristics, accordingly (Ruble and Martin 1998). Since these gender norms provide a solid picture of the family as a natural institution, with two parenting figures playing different and nearly complimentary duties, sexual minorities are regarded as a

danger to their ideology. In a number of countries, as in the case of Italy, gender-specific responsibilities appear to be embedded because individuals are socialized to gender roles at an early age via gender socialization (Best and Luvender 2015; Ruble and Martin 1998). Women are socialized to nurturing tasks such as sweeping and cooking or taking care of children, whereas men are socialized to dominance and power in multiple areas of both life and work, aspiring to management positions and upholding the masculine norm (Pacilli 2020).

These implicit social boundaries suggest that women and men are differently equipped, and this separation of roles appear to be grounded in human biology: this tendency to classify the social reality of gender traits is referred to as gender essentialism. Medin and Ortony (1989) introduced the term essentialism to explain the "inclination to perceive social categories as natural classes whose constituents share a common sense" and whose membership in each category is fixed and irreversible. The concept of essentialism holds that all members belonging within a category are likely to share similar distinct permanent features (Haslam et al. 2000). Therefore, essentialist theories' core element is to preserve societal inequities as inevitable, which promotes the status quo (Haslam et al. 2002). Essentialist viewpoints have frequently been used to justify prejudice against other groups (Gould 1981) and contribute to the gender gap because, as a logical outcome of biological differences, they aim even more to polarize the gender divide and to strengthen gender stereotypes (Bem 1993; Di Battista et al. 2022; Kay and Jost 2003; Pacilli et al. 2017). Such essentialist beliefs appear to be rooted mostly in individuals with conservative ideologies, as proponents of strict traditionalism emphasize the need for both paternal and maternal figures, relegating parenting to a binary division of roles and conveying the idea that a proper family structure should prevail (Ching and Wu 2022). For example, Pacilli et al. (2017) showed a strong association between endorsing an essentialist view about gender and the evaluation of same-sex parenting as unnatural, which then led to a lower evaluation of parenting competence.

The literature has already shown that those who condemn same-sex unions and parenting are motivated by fear of disobeying gender-based standards (Di Battista et al. 2020a, 2020b). Scali et al. (2014) examined the various correlates toward same-sex marriage and parenting and showed how being a woman, a low level of religiosity, left-wing political ideology, a high degree of education, and high socioeconomic position are correlated with more favorable attitudes toward them. On the contrary, a right-wing political stance appeared to be significantly associated with negative attitudes toward same-sex parenting as well as the recognition of a same-sex parent family (Trub et al. 2016). Stereotypical beliefs regarding same-sex parenting are largely influenced by hostility toward gay and lesbian persons, which is referred to as sexual stigma (Herek 2009a). Reactions towards sexual minorities are negatively impacted by the acceptance and application of sexual stigma, which is known as sexual prejudice (Herek 2009a). The heterosexism perspective, which continues to hold that heterosexual orientation represents the only available disposition (Herek 2007), has the potential to shape our perception of the world, influencing our decisions and leading to the rationalization and normalization of discriminatory social behaviors.

Individuals who hold sexual prejudice are more inclined to maintain anti-traditional gender beliefs and to be right-wing oriented, as the two fundamental precepts of conservatism are aversion to change and resistance to equality (Herek 2000; Jost 2006; Jost et al. 2003; Costa et al. 2014; Van der Toorn et al. 2017). Pacilli et al. (2011) found a relation between attitudes toward same-sex parenting and political affiliation, showing that those who identified with the right-wing position were more inclined to denigrate same-sex parents and consider them less competent.

### 1.2. The Present Research

The research described above consistently showed that a plethora of factors might be involved in the evaluation of same-sex parenting. In the present paper, we aimed

to contribute to this literature by investigating the association between the attribution of conflict within same-sex couples and the attribution of (less) parental competence, considering also the moderating role of political orientation. The current study is part of a growing body of research on people's attitudes toward same-sex parenting, which is a relevant and underexplored topic in the Italian context. To date, there is limited empirical research evidence to support the notion of a strategic use of conflict by opponents of same-sex parent families to devaluate these non-traditional families. However, research on separating same-sex parent families provides interesting insights to support our hypothesis on this strategic use of conflict. Separated same-sex parents often feel that they have failed to uphold the implicit expectation for same-sex parent families to be perfect in order to be socially accepted, leading to feelings of guilt and isolation (Gahan 2018). As such, in order to avoid the stereotype that same-sex couples are inferior and unable to maintain stable and lasting relationships compared with heterosexual couples, a lack of communication emerges which makes it impossible for conflict to be shown.

We investigated attitudes toward heterosexual and same-sex parents, focusing on a relatively unexplored aspect: the alleged attribution of conflict within same-sex couples. We assumed that perceivers may differentially link the attribution of conflict to parenting competence according to their political beliefs and the sexual orientation of the couple they evaluate. In particular, the main purpose of this exploratory study was to examine the relationship between the attribution of conflict and parenting competence as a function of (a) couples' sexual orientation, (b) participants' political affiliation, and (c) fundamental beliefs about gender essentialism and homonegativity. In accordance with the rationale stated above, our research aimed to investigate the following hypotheses: Regarding the evaluation of same-sex couples, we expected a negative association between the attribution of conflict and the evaluation of parenting competence (H1a); in the case of the evaluation of heterosexual couples, we expected that this relationship would be negative, but less intense (H1b). Furthermore, we expected that the attribution of conflict and the evaluation of parenting competence in same-sex couples would be significantly linked to the political orientation, the beliefs of gender essentialism and the level of homonegativity of the participants (H2a); in relation to the evaluation of heterosexual couples, we expected that these relationships would be not significant (H2b). Finally, only with regards to the same-sex couples, we explored whether the relation between the endorsement of gender essentialism and evaluation of parenting competence is mediated by the attribution of conflict: We expect this mediation to occur only among participants who endorse a right-wing political orientation (moderated mediation hypothesis: Hp3; see Figure 1 for the theoretical model).

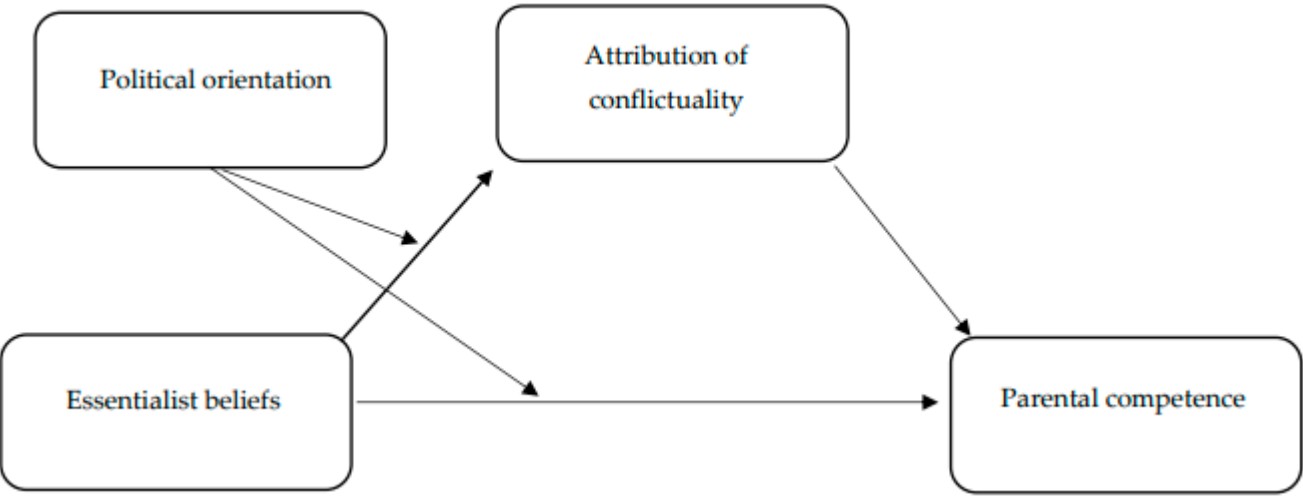

**Figure 1.** Theoretical model of the expectation of the evaluation of same-sex couples.

## 2. Methods

### 2.1. Participants

Before taking part in the study, participants were made aware of relevant aspects of the research, such as the research methods used and the affiliations of the researchers, in accordance with the Helsinki Declaration's ethical standards of 1964. After learning the instructions, participants confirmed that they had understood them correctly, agreed to participate, and declared to be of age. Four hundred thirty-six participants took part in the study (322 women, 106 men, and 8 non-binary). A technical error occurred and the online survey did not allow us to assess the participants' age. The participants self-reported their sexual orientation: three-hundred and eighty-three identified themselves as straight, 14 as gay and lesbian, 30 as bisexual, and 9 preferred to not declare. Participants were mainly recruited via snowball sampling through announcements on the authors' personal social media platform. Their involvement in the study was entirely voluntary and they were informed of their right to withdraw consent at any time. Participants filled in a questionnaire implemented using Qualtrics platform. The research, conducted in collaboration with the universities of Perugia, Bari, and RomaTre, was approved by the RomaTre ethical committee, where one of the authors was employed at the time of the data collection (April–May 2020).

### 2.2. Procedure

We presented participants with two fictitious scenarios in sequence (see the supplementary online materials). They were instructed to focus to these situations that can occur in real life. The scenarios depicted two distinct situations: One involving a same-sex couple and the other involving a heterosexual couple. For instance, the scenario concerning the same-sex couple stated: "Paolo and Michele have been civilly united for 4 years, after a 2-year engagement. The couple lives in a medium-sized town in central Italy". We deliberately avoided providing any other relevant information about couples in order to maximize the likelihood that any differences in the participants' assessments were due to the couples' own sexual orientation. After reading the different scenario, we asked participants to evaluate the same-sex couple and the heterosexual couple on a number of indicators. The order of the two scenarios was randomized and it did not produce differences in the results.

### 2.3. Measures

**Attribution of conflict.** We designed an ad-hoc scale for the purpose of the study. This scale consisted of 22 items that describe hypothetical events that can occur in real life and comprise several examples of conflictual behaviors (see the Supplementary Online Materials). Participants were asked to rate the likelihood that each situation can occur, in regard to the couple (heterosexual vs. same-sex) depicted in the recently read scenario. For example, participants were asked about the likelihood "that one partner may verbally abuse the other "or "that one partner feels less valued than the other at work". Responses ranged from 1 (=very unlikely) to 9 (=extremely likely), so high scores reflect a feeling of intense conflict. The reliability of the scale was optimal in relation to both the heterosexual ($\alpha = 0.94$) and same-sex ($\alpha = 0.92$) evaluation.

**Attitudes toward parenting competence.** Participants then provided their response on a series of statements on a scale of 1 (=absolutely not) to 9 (=absolutely yes) for the attribution of parental skills (Pacilli et al. 2011). The higher the score, the more favorable the evaluation of parenting. Examples of items are: "Paolo and Michele would be able to take care of their children adequately" or "Matteo and Lucia would be able to provide their children with good social rules" ($\alpha_{heterosexual} = 0.66$; $\alpha_{same\text{-}sex} = 0.67$).

**Homonegativity.** The Homonegativity Scale (HS) (Morrison et al. 1999; see for the Italian version (Faraci and Valenti 2020)) was adopted to detect homonegativity, defined as a set of beliefs hostile to gay/lesbian orientation, and consisted of six items and nine degrees, from 1 (=strongly disagreeing) to 9 (=strongly agreeing). For example, the participants

had to report whether they agreed with the following statements: "Homosexuality is not a mental disorder," "Homosexuals are immoral" ($\alpha$ = 0.70).

**Gender essentialism.** Participants completed the Gender Theory Questionnaire (Coleman and Hong 2008) in order to assess participants' level of gender essentialism, which is defined as the attitude of viewing social categories as natural qualities, such that men and women are endowed with different attributes that are considered natural, stable, and immovable (Bastian and Haslam 2006; Gelman 2003; Rothbart and Taylor 1992). Participants were required to indicate their level of agreement with the scale, which consisted of 11 items to evaluate whether gender differences are socially produced or determined by biology. Responses ranged from 1 (completely disagree) to 9 (completely agree). For example, "being male or female largely determines people's abilities and traits" or "as social contexts change, the characteristics we attribute to men and women are most likely to change" ($\alpha$ = 0.80).

**Political orientation.** Participants were asked to indicate their political orientation on a scale of 1 (left/liberal) to 9 (right/conservative). Higher scores therefore indicated adherence to right-wing political position.

### 3. Results

In order to verify our hypotheses, we first investigated the associations between the main variables. Correlations among the study variables are reported in Table 1. Relevant to the purpose of this study, same-sex couples' attribution of conflict was negatively related to the evaluation of same-sex parenting. In the evaluation of the heterosexual couple, this relationship appeared negative but less intense, confirming our hypothesis (H1a; H1b). Furthermore, in line with our predictions (H2a; H2b), the attribution of conflict in same-sex couples was positively associated with a right-wing political orientation, homonegativity, and gender essentialism; whereas the attribution of conflict among the heterosexual couple was unrelated to participants' political orientation and endorsement of gender essentialism.

**Table 1.** Means, standard deviations, and correlations between the study variables.

| | M | SD | 1 | 2 | 3 | 4 | 5 | 6 |
|---|---|---|---|---|---|---|---|---|
| **1. Attribution of conflict within heterosexual couple** | 4.55 | 1.28 | – | | | | | |
| **2. Heterosexual parental competence** | 6.29 | 1.41 | −0.295 *** | – | | | | |
| **3. Attribution of conflict within same-sex couple** | 4.24 | 1.19 | 0.764 *** | −0.244 *** | – | | | |
| **4. Same-sex parental competence** | 6.45 | 1.56 | −0.171 *** | 0.562 *** | −0.340 *** | – | | |
| **5. Political orientation** | 3.76 | 1.86 | −0.013 | −0.002 | 0.131 ** | −0.251 *** | – | |
| **6. Essentialism** | 3.61 | 1.30 | −0.027 | 0.082 | 0.131 ** | −0.194 *** | 0.377 *** | – |
| **7. Homonegativism** | 1.67 | 1.14 | 0.018 | −.078 | 0.238 *** | −0.352 *** | 0.398 *** | 0.401 *** |

Note. ** $p$ < 0.01, *** $p$ < 0.001.

*Moderated Mediation Analysis*

We then focused on the evaluation of the same-sex couple and tested a moderated mediation model (model 8 of the PROCESS macro; Hayes 2017) in which essentialism was modelled as a predictor, evaluation of parenting competence was modelled as a dependent variable, attribution of conflict to same-sex couples was considered as a mediator, and participants' political orientation was considered as a continuous moderator.

First, we ascertained that gender essentialism by political orientation interaction was a reliable predictor of attribution of conflict (b = 0.05, $p$ = 0.002; 95% CI: LL = 0.0201; UL: 0.0962): In particular, the effect of essentialism on attribution of conflict was significant only for participants with right-wing political orientation (b = 0.14; 95% CI: LL = 0.0457; UL: 0.2429) while it was not significant for left wingers (b = −0.03; 95% CI: LL = −0.1492; UL: 0.0889) or center wingers (b = 0.08; 95% CI: LL = −0.0051; UL: 0.1774).

We then confirmed that gender essentialism by political orientation interaction was a reliable predictor of evaluation of parenting competence (b = −0.10, $p$ < 0.001; 95% CI: LL = −0.1469; UL: −0.0545): In particular, the effect of gender essentialism on the evaluation of parenting competence was significant only for participants with right-wing political orientation (b = −0.21; 95% CI: LL = −0.3259; UL: −0.0865) while it was not significant for

left wingers (b = 0.09; 95% CI: LL = −0.0473; UL: 0.2390) or center wingers (b = −0.10; 95% CI: LL = −0.2156; UL: 0.0046).

Finally, we found that the indirect effect of gender essentialism on evaluation of parenting competence through the attribution of conflict was only significant for right-wing participants (b = −0.05; 95% CI: LL = −0.0948; UL: −0.0080) while it was not significant for left wingers (b = 0.01; 95% CI: LL = −0.0301; UL: 0.0573) or center wingers (b = −0.03; 95% CI: LL = −0.0670; UL: 0.0054). The index of moderated mediation was significant (b = −0.02; 95% CI: LL = −0.0372; UL: −0.0033). The findings confirmed our hypothesis (Hp3).

## 4. Discussion

Many countries are debating LGBT people's rights, including the right to wed and build a family. When it comes to the quality of parenting by gay and lesbian people, mainly negative attitudes persist, despite research showing that there is no reason to deny same-sex couples the prerogative to be parents (Johnson and O'Connor 2002; American Psychological Association 2004). Numerous studies have revealed that the mostly used standard to define family is a nuclear family (Barnard and Corrales 1979), with this structure being regarded as regular in contrast to other family structures that have been deemed dysfunctional and pathological. In order to reduce prejudice against same-sex parent families, a view of idealized same-sex relationships and families has often been promoted with several challenges regarding same-sex parent families remaining under-discussed, including parental conflict. As a consequence, individuals or families who fail to meet these idealized expectations and standards may face stigma and discrimination (Gahan 2018). Consequently, a vicious cycle a lack of communication develops in order to avoid the stigma that they cannot maintain a relationship as healthy and peaceful as heterosexual parents, same-sex parents may adhere to the heterosexist cultural bias that makes conflict appear to be a legitimate dimension only in heterosexual families.

In the present research, we aimed to examine the association between the attribution of conflict in same-sex couples and the attribution of minor parental competences. In particular, we focused on the attribution of conflict within the heterosexual couples and same-sex couples in order to investigate whether the alleged conflict attributed to same-sex couples vs. heterosexual couples can be used in strategic way to justify reduced same-sex parenting competence. As expected, the attribution of conflict leads to attribute less parental competence, and this relationship appears most pronounced especially in the evaluation of same-sex couples. Moreover, the attribution of conflict seems to be linked to right-wing political position, gender essentialist beliefs, and the level of homonegativity. Moderated mediation analyses supported our hypothesis, demonstrating that same-sex couples were perceived as less competent as parents by the attribution of conflict within the couple, particularly among participants with a high level of political orientation, i.e., right wingers.

In terms of political orientation, we assessed political tendencies through self-reported political positioning, which implies the skill to position one's own within two opposing political camps, that is the right wing and the left wing. We decide to use the right-wing and left-wing labels to connote the conservative and liberal standpoints, respectively. Future research should consider this aspect related to the political self-positioning, emphasizing the dimension of conservatism and progressivism, and focusing on the most predictive values of conservative political positions.

Moreover, due to the correlational nature of the research, it is impossible to conclude a causal relationship between the variables. To obtain causal inferences, we should conduct an ad-hoc experimental investigation that can clarify the direction of the emphasized link. In addition, the sampling technique may be a limitation of the research provided here. The snowball procedure cannot produce a representative sample of the entire reference population for obvious reasons. Nonetheless, historical circumstances associated with the

Sars-Cov-2 epidemic required the employment of this sampling method to obtain study participants. Future research can focus on representative sampling of the population.

Future research can also be conducted to deepen the comprehension of the attribution of conflict measure created ad-hoc for this study. In particular, future studies can address whether it is related to other partially overlapping instruments, such as for instance the conflict tactics scale, and they can also take into account other measures of attribution of conflict to investigate whether attribution of conflict and attribution of intimate partner violence are distinct dimensions (Ha et al. 2019).

In addition, this study compared two ad-hoc scenarios, one involving a straight couple and another considering a same-sex couple. Due to the lack of research on this subject, the focus of this study was initially placed on the gay male scenario. Further research can explore different scenarios involving other partners' configurations, such as of a lesbian couple. In relation to the evaluation of a lesbian scenario, for instance, two competing predictions can be made. On one hand, we can predict the same pattern when evaluating the parenting abilities as the one reported in our study due to negative stereotypes and biases related to sexual orientation that can impact their evaluation as parents. These biases stem from the societal expectations and the heteronormative statements surrounding the notion of family (Hopkins et al. 2013), which according to these mandates, must consist of two figures playing complementary roles, in order to be considered ideal. On the other hand, we can expect a distinct pattern due to the presence of two distinct nurturing figures. Allen and Goldberg (2020) emphasize the difficulties that lesbian women face in challenging gendered, heteronormative narratives about parenthood and marriage. In fact, prejudices persist despite the increasing amount of research that has highlighted the effectiveness of parenting in same-sex couples. This underlines the need to examine bias against both gay fathers and lesbian mothers in issues of same-sex parenting discrimination.

However, the study showed how political ideologies and conservative tendencies play a major role in shaping public opinion. Right-wing political stances and religious institutions appear to support an ideology that condemns same-sex relationships and encourage the dissemination of frequently homophobic messages that are capable of inciting extremely prejudiced behavior. It is important to remember that individuals then internalize these messages (Herek 2009b), revealing the enormous impact of the propagation of negative and obsolete belief systems. The current study and future research are meant to act as a starting point for the creation of interventions aimed at normalizing same-sex marriage and adoption, promoting precise and reliable information on sexual orientation, gender identity, and potential same-sex configurations, alongside new cultural models that can supplant beliefs anchored in a backward view.

**Supplementary Materials:** The following supporting information can be downloaded at: https://www.mdpi.com/article/10.3390/socsci12030128/s1, Scenarios adopted in the study.

**Author Contributions:** C.B., M.G.P. and S.P. conceived the original idea of this paper. A.T. and D.P. contributed to develop the original idea. C.B., M.T. and S.P. collected and analyzed data. C.B. drafted the first version of the paper. All the authors commented and revised the draft and contributed to finalize the paper. All authors have read and agreed to the published version of the manuscript.

**Funding:** This research received no external funding.

**Institutional Review Board Statement:** The study was conducted in accordance with the Declaration of Helsinki and approved by the Ethics Committee of University of Rome—RomaTre, where Dr. Daniele Paolini was employed at the time of data collection.

**Informed Consent Statement:** Before taking part in the study, individuals were made aware of some aspects of the research, such as the research methods used and the affiliations of the researchers, in accordance with the Helsinki Declaration's ethical standards of 1964. After learning the instructions, participants confirmed that they had understood the instructions correctly, agreed to participate, and declared to be of age. Their involvement in the study was entirely voluntary and anonymous.

**Data Availability Statement:** Data are available upon request to the corresponding author.

**Conflicts of Interest:** The authors declare no conflict of interest of data.

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
