# Peer review of "Same-Sex Parenting Competence Evaluation: The Role of Gender Essentialism, Political Orientation, and Attribution of Conflict"

_socsci, doi:10.3390/socsci12030128_

Round 1

Reviewer 1 Report

I really enjoyed reading this manuscript, which deals with a socially controversial issue that still needs scientific attention. The writing is clear, the literature review is well documented and the rationale behind the idea and the hypotheses is straightforward and well explained. I also found the idea of analyzing the role of perceived conflict very original and compelling. Therefore, I have only minor observations.

Although the idea of focusing on the mediational role of perceived conflict is original and intriguing (as far as I know no study has so far examined this issue), I still think that a rationale or at least a reasoning should be provided on why the attribution of conflict was chosen over other variables.

Hyp 3 is not completely clear to me. The authors wrote “Finally, we explored whether the relation between the endorsement of gender essentialism and attribution of parenting competence to same-sex (vs. heterosexual) couples is mediated by the perception of conflict”. The word “vs” leads the reader to think that the same- and different-sex couples would be compared. I found out that this was not the aim of the study only when I read the Results section. I suggest reformulating this hypothesis.

Related to that, it is not completely clear to me whether the authors’ intention was to run two models (one for heterosexual and one for homosexual couples), but the results of the correlations changed the plans to test only the model for homosexual couples or whether the intentions was to test only one model since the beginning. This should be made clear.    

Given that, in Italy, the political spectrum is defined by left and right orientation (and not by liberals and conservatives), I would stay with this distinction and avoid referring to liberals or conservatives (especially in the hypotheses and in the method section). This parallelism might be suggested and discussed in the GD.

I found it difficult to read the results when political orientation was expressed in terms of high-medium-low levels. I suggest keeping only the use of left, right, and centre which makes the results more readable.

I think that it might be useful providing a rationale for why only male homosexual couples participated in the study (beyond heterosexual ones). The authors have already discussed plausible differences in the results with lesbian couples, but they did not justify their choice in the presentation of the study. I guess that the reason for this choice could have been that male same-sex couples are generally considered less suited to raise children than female same-sex couples, but of course, I leave it to the author to explain their choice.  It would also be plausible that lesbian couples would be considered less suited to raise children than heterosexual couples for different reasons than attribution of conflict, for the reasons explained at p. 8.

p. 8 line 305 there is an error in the sentence

Author Response

Please find the answers to your concerns in the attached cover letter

Reviewer 2 Report

Thank you for the opportunity to review the article entitled “Same-sex Parenting Competence Evaluation: The role of Gender Essentialism, Political Orientation, and Attribution of Conflict”. This study examined the relations of perceived parental competence of same-sex parent families to attribution to conflicts, gender essentialism, and political orientation in an Italian context. Whereas the experimental design of this study is commendable, I think that the current manuscript can be improved in several ways. Below are some considerations for the authors and editor.

First, the authors may review more literature on their key constructs “attribution to conflict” and “perceived parental competence”. In the current manuscript, the definition of “attribution of conflict” is unclear (sometimes the term “perception of conflict” was used interchangeable with “attribution of conflict”, it sounds confusing, I suggest that the authors standardize the use of the terms for key variables). Because of the lack of literature review and clear definition of this construct, I found it difficult to understand the hypotheses on p.4. When I investigated the measure of “attribution of conflict” on p.5, I was further confused because it was about the “likelihood that, for example one partner may verbally abuse the other”. How does this measure reflect “attribution”? I believe that another term for this construct should be used instead of “attribution of conflict”.

Second, the literature of the relations of perceived parental competence to gender essentialism, political orientation, and “attribution of conflicts” was lacking. The authors have reviewed some studies on people’s biases or perceptions about the parental competence of same-sex parents, whereas this study was not just about perceived parental competence of same-sex parents, but also its relations to other variables. Therefore, I suggest that the authors also incorporate this important part of literature review in their revised version. The same applies for the relations of “attribution of conflict” to gender essentialism and political orientation.

Third, how the mediation and moderation hypotheses came about was also unclear. On what basis did the authors speculate that “attribution of conflict” would mediate the association between gender essentialism and perceived parental competence in the same-sex parent scenario? What is the theoretical reason behind this hypothesis? Is it possible that perceived parental competence serves as a mediator between gender essentialism and attribution of conflict? Similarly, why did the authors hypothesize that political orientation would moderate the associations between gender essentialism and attribution of conflict and between gender essentialism and perceived parental competence? Is it possible that gender essentialism is a moderator for the connection between political orientation and attribution of conflict and between political orientation and perceived parental competence? In the current version of the manuscript, theoretical foundations or arguments for these hypotheses are lacking.

Fourth, I wonder why the words “prototypical” and “non-prototypical” were used. I think the terms “heterosexual” and “same-sex” parents are fine.

Fifth, the authors may read and cite the following recent article on individuals’ perceptions of same-sex parent families that is highly related to their current work.

“Ching, B. H.-H. & Wu, H. X. (2022). Ideological beliefs and gender essentialism: Relations to individual and normative opposition to same-sex parent families. Psychology and Sexuality. Advance online publication. https://doi.org/10.1080/19419899.2022.2075789”

Finally, the authors only recited their findings in the Discussion without elaboration and proceeded to several limitations of their study and future directions. I would like to see a more in-depth Discussion of their findings in a revised version of their manuscript.  

Author Response

(The authors gave the same response as above.)

Reviewer 3 Report

- The authors studied the perception of conflict within heterosexual/prototypical couples and same-sex/non-prototypical couples, to investigate whether the alleged conflict attributed to same-sex couples vs. heterosexual couples could be used in strategic way to justify reduced same-sex parenting competence.

- The topic is novel and interesting.

- The paper is well structured. There are good figures and tables.

- The Discussion is too short in the current format and needs to be expanded citing more articles related to the topic of your work. For example, cite and comment the following article by Costa D. (2022). The influence of social capital on health issues among transgender and gender diverse people: a rapid review. Science & Philosophy, 10(2), 109-131.

In the discussion section the critical point of view of the authors must raise.

Author Response

(The authors gave the same response as above.)

Round 2

Reviewer 2 Report

I appreciate that the authors have addressed my concerns adequately. 

Author Response

Dear rewiever,

we are very glad that you a appreciated our revision in response to your comments.